# The Invasive Alien Species *Callinectes sapidus* Threatens the Restoration of *Ostrea edulis* and *Paracentrotus lividus* in the Mediterranean Sea

**DOI:** 10.3390/ani15243553

**Published:** 2025-12-10

**Authors:** Gianni Brundu, Philip Graham, Mattia Corrias, Cheoma Frongia, Stefano Carboni

**Affiliations:** 1IMC—International Marine Centre, Loc. Sa Mardini, Torre Grande, 09170 Oristano, Italy; p.graham@fondazioneimc.it (P.G.); m.corrias@fondazioneimc.it (M.C.); c.frongia@fondazioneimc.it (C.F.); s.carboni@fondazioneimc.it (S.C.); 2NBFC—National Biodiversity Future Center, 90133 Palermo, Italy; 3Department of Comparative Biomedicine and Food Science, University of Padova, Viale dell’Università 16, Legnaro, 35020 Padova, Italy

**Keywords:** European flat oyster, sea urchin, blue crab, predation, alien species

## Abstract

The blue crab *Callinectes sapidus* is an invasive species in the Mediterranean Sea, raising concerns for biodiversity and restoration projects of native species. In this study, we examined whether blue crab preys on two native species with ecological and commercial importance, the European flat oyster *Ostrea edulis* and the sea urchin *Paracentrotus lividus*. By testing different prey sizes, we found that the blue crab strongly prefers small oysters, while sea urchins were rarely eaten. Medium and large individuals of *O. edulis* were resistant to predation. These results indicate that the blue crab may threaten restoration efforts that rely on releasing small hatchery-reared oysters, which are highly vulnerable. Considering predator presence and applying suitable management strategies will be essential to improve the success of restoration programs in the Mediterranean Sea.

## 1. Introduction

The European flat oyster *Ostrea edulis* and the stony sea urchin *Paracentrotus lividus* play important roles in marine ecosystems, as they are strongly connected with other species in the food web. *Ostrea edulis* is an ecosystem engineer that creates biogenic reefs supporting complex communities [1], and *P. lividus* is a herbivorous keystone species central to the “fish–sea urchin–macrophyte” tri-trophic interaction [2]. This implies that modifications in the abundance and size structure of their natural populations can trigger detrimental trophic cascading effects, potentially leading to habitat degradation and the loss of related ecosystem services. Therefore, effective strategies to manage and conserve wild stocks are mandatory.

In Europe, *O. edulis* and *P. lividus* have been commercially important resources for centuries, and, in some countries, they have been overexploited [3,4]. The high fishing effort, as well as disease, pollution and climate change, contributed to the depletion of natural populations and the functional extinction of *O. edulis* in most European regions [3,5,6], as well as to a sharp decline in the abundance of *P. lividus* in several countries, including North Brittany [7], Ireland during 20 years of intensive fishing [8], Marseille coasts in France and Sardinia Island in Italy [9].

Nowadays, the harvesting of *O. edulis* and *P. lividus* is regulated or limited by international, national and local legislation. *O. edulis* is included on the list of threatened and/or declining species and habitats of the Oslo-Paris Commission [10] and EU Habitats Directive [11,12], and is the focus of several national and international efforts to overcome existing barriers to the responsible conservation, restoration, and recovery of the species [13]. The harvesting of *P. lividus* is regulated by national law in the Maltese Islands (L.N. 149 of 2023, https://legislation.mt/eli/ln/2023/149/eng, accessed on 6 December 2025), as well as by regional laws or local decrees in France (Arrêté n. R93-2023-09-29-00001, https://www.dirm.mediterranee.developpement-durable.gouv.fr/periodes-interdites-pour-la-peche-des-oursins-en-a34.html?lang=fr, accessed on 6 December 2025), Spain (https://miprincipado.asturias.es/bopa/2020/09/29/2020-07773.pdf, accessed on 6 December 2025) and Italy [14].

However, harvesting limitations and regulations are often ineffective or insufficient to ensure the conservation and recovery of wild stocks, largely due to the overcapacity of fishing fleets, the inadequate consideration of ecosystem-level effects of fishing, and the lack of enforcement of unpopular but necessary reductions in fishing effort [15]. In such cases, ecological restoration through the release of hatchery-reared individuals is increasingly recognized as a viable strategy to recover degraded marine ecosystems and rebuild functionally extinct populations [16], including *O. edulis* [17,18] and *P. lividus* [19,20,21].

Although positive results have been previously obtained in restocking programs of marine species [22], the effectiveness of this practice must be carefully evaluated, as the use of captive-born individuals often results in high mortality rates after release into the wild [23], with predation being among the primary causes [22,24]. The risks posed by predation in particular can be overlooked when planning and executing restoration projects, especially when new alien species have recently and sometimes prominently made their appearance in a new region [25]. Invasive alien species are increasingly recognized as one of the main barriers to conservation and restoration of marine biodiversity in Europe and are one of the key drivers of biodiversity loss worldwide [26].

The blue crab *Callinectes sapidus* is an invasive alien species accidentally introduced into the Mediterranean Sea in 1948 [27,28]. It is a major predator of the oyster *Crassostrea virginica* [29] and is considered a generalist, opportunistic omnivorous [30], able to feed on a variety of food resources, including molluscs, crustaceans, and finfish [31]. Recently, the abundance of the blue crab has gradually increased in the Mediterranean basin, mainly in the Adriatic Sea, raising concerns about the ecological impact of its introduction and becoming a threat to overall biodiversity, local fisheries [32,33,34] and commercial farmed species like Mediterranean mussel (*Mytilus galloprovincialis*), Pacific cupped oyster (*Magallana gigas*, formerly *Crassostrea gigas*) and Japanese carpet shell (*Ruditapes philippinarum*) [35]. Shauer et al. [36] examined the invasion dynamics of *C. sapidus* across the Adriatic, Ionian, and Central Mediterranean subregions, identifying an early introduction period from 1965 to 1999, followed by an establishment phase from 2000 to 2015, and a rapid expansion phase from 2016 to 2024. The annual CPUE (Kg wet mass^−1^ boat-year^−1^) of *C. sapidus* in the Adriatic Sea remained consistently below 1.5 until 2017. Subsequently, change point analysis revealed a significant shift in CPUE trends, with values exceeding 9000 in 2023 [37].

Although several restoration projects targeting *O. edulis* and *P. lividus* have recently been developed in the Mediterranean Sea [21,38], the potential predation of *C. sapidus* on these species has so far received little attention, especially when considering the introduction of captive-bred individuals for restoration purposes. The aim of this study was to evaluate the predation rate of *C. sapidus* on three size classes (small, medium, and large) of *O. edulis* and *P. lividus*. We hypothesised that the blue crab prefers to feed on smaller individuals of both species, potentially representing a limiting factor during restoration and pointing out possible negative consequences on wild populations’ natural recruitment. In fact, as observed in previous restoration attempts, many invasive non-native species can negatively impact the conservation objectives for protected species and habitats, as well as the associated biodiversity [39].

## 2. Materials and Methods

### 2.1. Experimental Design

A total of 18 predation experiments were carried out over a two-month period (October–November 2023), combining two predator sexes (male and female *C. sapidus*) with nine prey treatments (species × size class): small *O. edulis*; medium *O. edulis*; large *O. edulis*; small *P. lividus*; medium *P. lividus*; large *P. lividus*; a mix of small *O. edulis* and small *P. lividus* (50:50, number of individuals); a mix of medium *O. edulis* and medium *P. lividus* (50:50, number of individuals); a mix of large *O. edulis* and large *P. lividus* (50:50, number of individuals) (Figure 1).

For each experiment, five replicate predators per sex were offered 10 prey items. Consumption was monitored at 1, 4, 8 and 24 h after prey placement, with experiments starting at 10:00 am, following the methods described in Prado et al. [40]. Predation rate was expressed as the percentage of prey consumed relative to the initial number offered [41].

All experiments were carried out in five 150 L gently aerated tanks (one tank per predator), at 37, 25 °C and a 14 h light/10 h dark photoperiod. Biomass (wet weight) and morphometric measurements of predators and prey were recorded (Table 1; Figure 2).

### 2.2. Predators

Wild male (*n* = 5) and female (*n* = 5) individuals of *C. sapidus* were manually collected at 1 m depth in the S’Ena Arrubia lagoon, Sardinia, Italy (39°49′43.90″ N, 8°33′14.39″ E). S’Ena Arrubia is a small (120 ha), brackish (from 0.9 to 39.5) and shallow lagoon (0.4 m mean depth), connected to the sea by a single artificial outlet [42]. Crabs were transported in thermally insulated and refrigerated containers to the facilities of the International Marine Centre (IMC), Sardinia, Italy. Upon arrival at the laboratory, all crabs were visually inspected to ensure that only individuals in excellent health and at the intermolt stage (e.g., hard carapace, absence of missing appendages or carapace damage) were included in the experiment. Crabs were also monitored for mortality or signs of impaired health throughout both the acclimation period and the experiment trials.

Predation by males and females was assessed separately, given sex-related differences in claw morphology that may result in distinct functional responses [29]. Each crab was tested across all prey species and size classes. Prior to each experiment, predators were starved for 24 h to ensure empty stomachs and standardized hunger levels [40,43].

### 2.3. Ostrea edulis Prey

Captive-born *O. edulis* originated from wild adults collected in the Nora lagoon, Sardinia, Italy (38°59′15.16″ N; 9°0′28.30″ E). Adults were transferred to the IMC hatchery and maintained in gently aerated 150 L tanks within a recirculating aquaculture system (RAS), operating at a flow rate of 2.5 L·min^−1^. Broodstock conditioning and larval production followed the methods described in Maneiro et al. [44] and Helm et al. [45], with a temperature gradient of 13–18 °C, a photoperiod gradient of 11–16 h light, and 37. Oysters were fed a mixed microalgal diet of *Isochrysis galbana* (50%) and *Chaetoceros gracilis* (50%), equal to 6% dry-weight algae per dry-weight oyster day^−1^ oyster^−1^.

Larvae and post-larvae were reared at 37 and 18 °C under gentle aeration and fed *I. galbana* (50 cells µL^−1^) and *C. gracilis* (50 cells µL^−1^). Settlement was induced by exposing larvae to oyster shell fragments (0.5–1 mm microcultch) [46].

Large and medium oysters were hatchery-produced in summer 2021 and 2022, respectively, and subsequently farmed in the Nora lagoon following the methods described in Brundu et al. [47]. Juvenile oysters (0.2 ± 0.1 g wet weight, 11 ± 0.2 mm shell length) were seeded at the end of June in floating bags attached to fully submerged ropes. Throughout the farming period, water temperature ranged from 11.9 °C in January to 30.1 °C in July, and the oysters relied exclusively on naturally occurring phytoplankton in the lagoon as their food source [47]. Small individuals were hatchery-produced in summer 2023.

### 2.4. Paracentrotus lividus Prey

Small individuals of *P. lividus* originated from mixed-broodstock adults (test diameter > 50 mm) collected at 5 m depth in Su Pallosu Bay, Sardinia, Italy (40°2′46.53″ N, 8°24′35.89″ E) in 2023.

Reproduction, larval, and post-larval rearing followed the methods described by Brundu et al. [48,49]. Larvae and post-larvae were reared at the IMC facilities in natural seawater at 37 and 20 °C, under a 14 h light/10 h dark photoperiod.

Individuals were maintained in a RAS (flow rate of 5 L·min^−1^) consisting of three rectangular 250 L tanks equipped with biological and mechanical filtration (10 µm). They were fed ad libitum with natural biofilm of diatoms and fresh thalli of *Ulva* sp. for approximately eight months.

For the experiments, medium and large individuals of *P. lividus* were collected by scuba diving from a wild population in Su Pallosu Bay, Sardinia, Italy (40°2′46.53″ N, 8°24′35.89″ E).

### 2.5. Compressive Strength

For each of the prey species and size classes, the shell breaking strength of ten randomly selected individuals was measured following the methods described in Newell et al. [43] for *O. edulis* and Asnaghi et al. [50] for *P. lividus*. Measurements were performed using the custom-built mechanical press (piston area 76.9 mm^2^) described by Asnaghi et al. [50].

*O. edulis* individuals were placed with the right flat valve on the press base and the cupped left valve facing upward [51]. *P. lividus* individuals were positioned with the oral surface facing downward, replicating their natural orientation in the wild.

Increasing pressure was applied through a piston until crushing occurred. The compressive force required to crush each specimen was recorded as the total weight (Kg) on the piston (Figure 3) and converted into newtons (N).

### 2.6. Data Analysis

Data were analyzed by Statistica 6.1 (StatSoft, Inc., Tulsa, OK, USA, 2004). Normality was tested with the Shapiro–Wilk test, and homogeneity of variances with Levene’s test. Since the assumption of normality was violated, non-parametric tests were applied. Differences among size categories (fixed factor, three levels: small, medium, and large) within each prey species were assessed with the Kruskal–Wallis test (*p* < 0.05), as a prerequisite to attribute size effects on predation. Differences between the sexes of predators (male and female) were investigated with a *t*-test (*p* < 0.05). A three factors General Linear Model (GLM) test was employed to investigate different predation between sex of predator (two levels, male and female), prey species (two levels, *O. edulis* and *P. lividus*) and size (three levels, small, medium, and large), at 1, 4, 8 and 24 h (*p* < 0.05), as well as prey hardness. Tukey’s Honestly Significant Difference (HSD) test was applied to evaluate all pair-wise treatment comparisons (*p* < 0.05).

## 3. Results

Males of *C. sapidus* resulted in higher biomass (*p* < 0.01) and claw (*p* < 0.001) than female individuals. Both *O. edulis* and *P. lividus* resulted in significant differences among the three size classes (small, medium, and large). In particular, the Kruskal–Wallis test evidenced differences for all variables investigated (Table 2).

### 3.1. Predation

Predation increased from 1 to 24 h of prey exposure and varied according to prey species, prey size, and predator sex. No predation was recorded on medium and large-sized individuals of either species. However, although no quantified predation attempts were observed on medium *O. edulis*, leaving visible shell breakages at the extremities (Figure 4).

Both female and male crabs recorded significantly higher predation (*p* < 0.001) on small than medium and large *O. edulis* when offered as a single prey species. In addition, females displayed significantly different predation rates among size classes of *P. lividus*, with higher predation (*p* < 0.01) on small than medium and large individuals from 1 to 24 h of exposure (Table 3). Comparison between prey species resulted in a higher predation (*p* < 0.001) on small *O. edulis* than on small *P. lividus* after 24 h of exposure. Predation on *O. edulis* reached 94 ± 4% (mean ± SE, females) and 62 ± 19% (males), compared with 14 ± 7% (females) and 0% (males) on *P. lividus* (Figure 5, Table 3).

Similar results were obtained when a mix of the two prey species was offered. Both male and female *C. sapidus* showed significantly higher predation on small than medium and large individuals of *O. edulis* (*p* < 0.001). After 24 h of exposure with mixed prey, predation was higher (*p* < 0.001) on *O. edulis* (68 ± 15% females; 92 ± 8% males) than on *P. lividus* (8 ± 5% females; 32 ± 12% males) (Figure 5, Table 4).

Regarding predator sex, females exhibited a significantly higher predation than males after 4, 8 and 24 h of exposure (*p* < 0.05), when prey were offered as a single species (Table 3). Conversely, males showed higher predation than females (*p* < 0.05) when prey were offered as a mix (Table 4).

### 3.2. Compressive Strength

The compressive force required to crush prey differed significantly among species and size classes. Medium *P. lividus* (101 ± 4.6 N) and small *O. edulis* (90.2 ± 12.3 N) resulted in similar strength values; both were significantly higher (*p* < 0.001) than small *P. lividus* (19.9 ± 1.5 N) but lower than large *P. lividus* (168.2 ± 9.4 N).

The compressive strength of medium and large *O. edulis* could not be measured, as their shells did not crush under the maximum pressure applied (500 N) (Figure 6).

## 4. Discussion

*C. sapidus* is one of the most extensively studied decapods in native habitats, and yet information on its feeding behavior in non-indigenous areas is limited [41]. To our knowledge, this is the first experimental evidence of predation by *C. sapidus* on *O. edulis* and *P. lividus* in the Mediterranean Sea. Previous reports include only a single field observation of predation on *O. edulis* in Greece [31].

Only low predation rates on small-sized individuals of *P. lividus* were recorded in this study, both when offered alone and in mixed diet with *O. edulis*. *C. sapidus* is an opportunistic omnivore consuming a wide range of prey, including decaying animals, conspecifics, macrophytes, algae, plants, detritus, as well as fish and invertebrates [52,53,54]. Therefore, a higher predation on *P. lividus* might have been expected. We hypothesize that the low predation observed was related to poor palatability rather than to mechanical resistance, since *P. lividus* exhibited a lower compressive strength than the widely predated *O. edulis*. In particular, we observed that the crabs showed limited engagement when handling sea urchins, suggesting that the difficulties imposed by their spines and pseudo-spherical body shape may act as an effective morphological deterrent.

Beyond physical and energetic constraints, the nutritional and palatability features of prey strongly influence prey selection by *C. sapidus* [55]. In this study, a clear feeding preference for *O. edulis* (>62% predation) over *P. lividus* (<32% predation) was recorded, mainly when offered as a mixed diet. Similar selectivity patterns have been reported in previous studies, such as the preferences of *C. sapidus* for *M. galloprovincialis* (71% predation) over *M. gigas* (8% predation) [40], or grooved carpet shell *Ruditapes decussatus* over *M. galloprovincialis* and lagoon cockle *Cerastoderma glaucum* [41].

It is important to note that our results were obtained under controlled laboratory conditions, where *O. edulis* and *P. lividus* were offered as the only available prey. In natural settings, however, these species constitute only a fraction of the potential prey spectrum accessible to *C. sapidus*. In addition, the burrowing behavior of the blue crab often favors the exploitation of infaunal bivalves, particularly clams [31,41]. This constraint in prey availability may therefore have influenced predation rates and prey selection observed in our experiments. In the wild, broader prey availability and the higher accessibility of certain bivalve species could lead to different feeding preferences, potentially reducing or shifting predation pressure on *O. edulis* and *P. lividus*.

No predation occurred on medium and large *O. edulis*, although attempts at predation and breakages at the shell extremities of medium-sized oysters were observed. This suggests that individuals with a shell length of ~60 mm (medium size) and larger are resistant to blue crab predation, confirming they can coexist with the presence of *C. sapidus*, as observed in the Ebro Delta [40].

Conversely, the high predation recorded on small *O. edulis* (~33 mm shell length) indicates that their shells are vulnerable to crab claws. This result contrasts with the findings of Eggleston [56], who found a critical crush size of ~30 mm for *C. virginica*. We hypothesize that the different crush resistance recorded for *O. edulis* (this study) and *C. virginica* [56] may be due to morphological differences between species, as previously observed between *C. virginica* and *C. ariakensis* [43,51,57]. This hypothesis is supported by the fact that we adopted experimental methodologies and crab sizes (135–165 mm CW) similar to those used by Eggleston [56].

The shell breaking strength of oyster species varies according to specific shell characteristics, in particular the foliated layer [51]. The shape and curvature of the shell can distribute mechanical load across the oyster [58]; however, the shell thickness and density are the main mechanical properties influencing the compressive force required to deform, crack and crush the shell, in turn affecting the predation rates [51]. This is confirmed by Newell et al. [43] and Bishop & Peterson [57], according to which a higher predation rate of *C. sapidus* on *C. ariakensis* than on *C. virginica* was due to a weaker shell of *C. ariakensis*.

Similarly to the results obtained in this study, a high predation rate on small oysters and low to negligible predation on larger individuals was reported in previous laboratory experiments. Eggleston [56] recorded a higher predation rate of adult *C. sapidus* (135–165 mm width) on small (15 mm shell length) than on medium (25 mm shell length) and large (35 mm shell length) individuals of *C. virginica*. Prado et al. [40] observed that adult crabs (~158 mm width) were able to prey on small (~70 mm shell length) individuals of *M. gigas*, but not on larger ones (>83 mm shell length). Similar results were obtained by Seed [59] and Hughes and Seed [60] with the mussel *Geukensia demisa*, showing that *C. sapidus* prefers small over large individuals, thereby minimizing handling time and maximizing net energy intake.

In contrast, predation on native Mediterranean bivalves resulted in a significant preference for large over medium-sized individuals of *R. decussatus*, suggesting that the potential energetic benefit of consuming larger individuals offsets the high shell strength and hardness [41]. Conversely, no size-based preferences were observed for *M. galloprovincialis* or *C. glaucum*, indicating that adult blue crabs are equally capable of feeding on both small (~28 mm shell length) and large (~61 mm shell length) individuals of *M. galloprovincialis* [40].

Previous studies observed distinct functional responses of male and female *C. sapidus* when preying on *C. virginica* across size classes and stocking densities, due to morphological dimorphism in chelae (greater crushing strength and larger chelae of males than females) [29,61]. In particular, Eggleston [29] reported a positive density-dependent functional response of large females, while males exhibited an inverse density-dependent response.

In our study, females exhibited higher predation in single-prey treatments, while males dominated in mixed diets. Differences in predation rates between sexes were recorded, but higher predation could not be consistently attributed to either sex: females exhibited higher predation when offered a single prey species, males when offered a mixed diet, and both sexes showed similar predation after 24 h of prey exposure. The absence of different sex-related predation is consistent with Ortega-Jiménez et al. [62], who found no diet-related differences between sexes in wild crabs.

The high predation recorded in this study on small-sized individuals of *O. edulis* indicates that *C. sapidus* represents a major threat to restoration success, especially regarding juvenile individuals below 60 mm in shell length, which are usually employed in restoration programs [63,64].

Predation is a key factor in determining the success or failure of restoration projects, as it represents a major cause of mortality in juvenile animals [65]. This is particularly true for hatchery-reared animals, which often exhibit deficits resulting from domestication [22,56]. One of the most important deficits is their inability to recognize predators or to apply appropriate anti-predator responses [22].

This deficit could be mitigated through pre-release conditioning and training aimed at eliciting anti-predator responses [22]. In the case of sessile bivalve species like *O. edulis*, increasing shell hardness or strengthening the adductor muscle may decrease predation rates [40]. In fact, Newell et al. [43] observed a significant strengthening of the shells of both *C. virginica* and *C. ariakensis* when exposed for two months to the presence of *C. sapidus*. In the case of motile species like *P. lividus*, exposure to predator presence or cues (i.e., chemical signals) could serve as suitable training, enabling them to recognize predators and seek shelter [19].

Alternatively, site selection can be fundamental for the success of restoration projects. Choosing locations with low predator abundance can improve survival, although this requires prior assessment of predator presence and abundance [24]. When predators cannot be avoided through site selection, it is essential to adopt alternative methodologies to limit predation pressure, such as adjusting the timing of seabed deployment or employing caging techniques [24].

Timing is one of the most critical factors affecting the survival of juveniles after release into the wild. It is fundamental to release the animals when environmental conditions (e.g., salinity, temperature, and oxygen) are suitable for survival and growth, and when food availability is high, phytoplankton for oysters [46] and macroalgae for sea urchins [14].

In the Mediterranean Sea, the most suitable period for the growth and survival of *O. edulis* is from June to November [47,65], similarly, spring to autumn for *P. lividus* [66].

However, it is also important to release the animals when predators are less active. In the case of *C. sapidus*, this corresponds to winter dormancy, when individuals are mostly buried and hibernating [67]. In particular, in Sardinia, temperatures are unsuitable for blue crab activity from November to April [68]. This suggests that late autumn (i.e., November) may be the most suitable period for the restoration of *O. edulis* and *P. lividus*, representing a compromise between favorable environmental conditions for released animals and reduced predation pressure of *C. sapidus*. The present experiment was conducted during October-November at a temperature (25 °C) similar to that recorded at the predator and prey collection sites, further supporting the suitability of this period for animal release in Sardinia. However, in other geographic areas with different climatic conditions (e.g., seawater temperature and salinity), it may be necessary to acclimate the animals to field conditions prior to their release into the wild.

Caging can further enhance survival by providing protection to the juvenile individuals until they reach the refuge size. Suspended cages are generally more protective than on-bottom cages against benthic predators [45], but their effectiveness against an agile swimmer like *C. sapidus* requires further testing. For benthic grazers such as *P. lividus*, on-bottom cages may be more appropriate, but rigid structures resistant to crab claws should be used in both cases.

These findings highlight how invasive alien predators can compromise the restoration goals of native species. Incorporating predator management into restoration strategies will be essential to ensure functional ecosystem recovery in the Mediterranean Sea.

## 5. Conclusions

*P. lividus* was not among the preferred prey species of the blue crab *C. sapidus* and appeared to be consumed only occasionally. In contrast, small individuals of the European flat oyster *O. edulis* were heavily preyed upon, suggesting that *C. sapidus* represents a major threat to the conservation and recovery of this species.

Medium-sized oysters (~60 mm shell length) showed resistance to blue crab predation, suggesting that this size class provides an effective mechanical refuge. However, future climate change scenarios may alter this threshold, as ocean acidification and warming can reduce shell strength, increasing vulnerability to crushing [69,70,71].

Given the increasing abundance of the blue crab across the Mediterranean Sea [34] and the ongoing restoration initiatives targeting *O. edulis* and *P. lividus*, our findings underscore the importance of evaluating the presence and abundance of *C. sapidus* at restoration sites. Where necessary, management actions such as pre-release conditioning and training of animals, site selection, timing of release, and the use of protective caging should be implemented to mitigate predation pressure on small-sized individuals and enhance restoration success (Figure 7).

This study provides the first experimental evidence of *C. sapidus* predation on the native species *O. edulis* and *P. lividus*, both targeted for restoration in the Mediterranean Sea. The findings offer practical guidance for improving the survival of hatchery-reared juveniles and enhancing the overall success of restoration programs. Incorporating predator assessment into restoration planning should become standard practice to strengthen the resilience and long-term effectiveness of Mediterranean restoration efforts.

## Figures and Tables

**Figure 1 animals-15-03553-f001:**
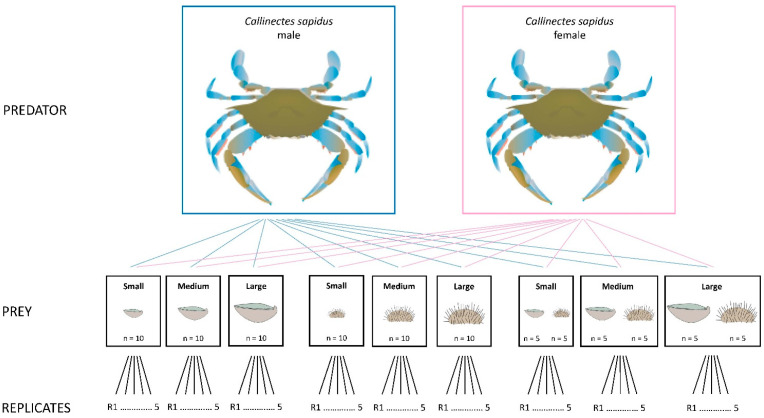
Experimental design to test the predation rate of male and female *Callinectes sapidus* (*n* = 5 per sex) on three size classes (small, medium, and large) of *Ostrea edulis* and *Paracentrotus lividus*, offered either as single prey species or as a mixed diet.

**Figure 2 animals-15-03553-f002:**
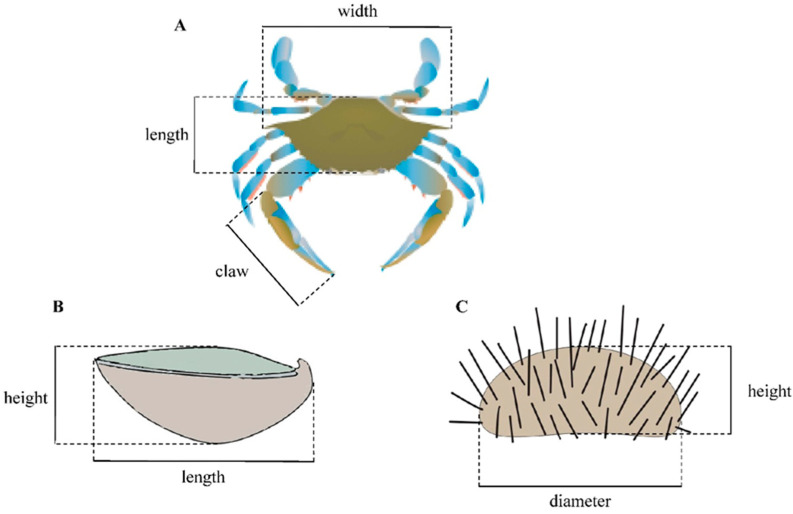
Biometric measurements of *Callinectes sapidus* (**A**), *Ostrea edulis* (**B**), and *Paracentrotus lividus* (**C**).

**Figure 3 animals-15-03553-f003:**
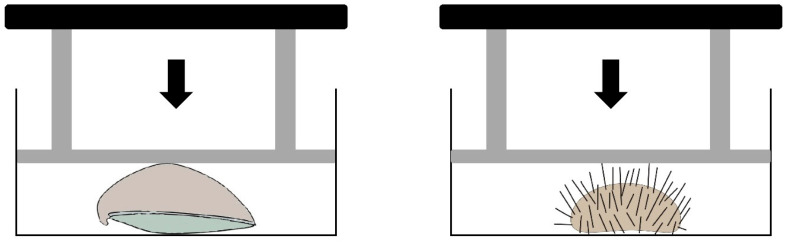
Custom-made mechanical press used to measure the compressive strength of prey species and size. Black arrows indicate the direction of applied force (modified from Asnaghi et al. [50]).

**Figure 4 animals-15-03553-f004:**
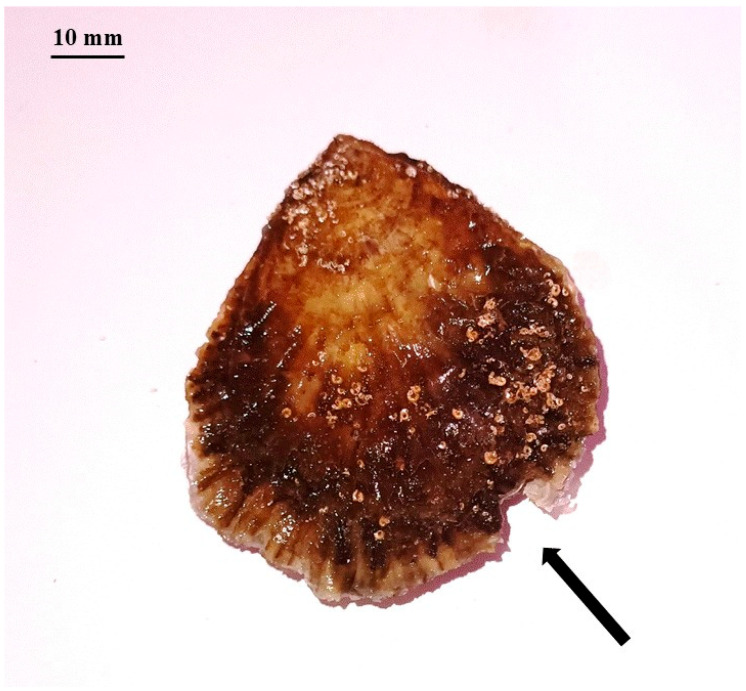
Medium-sized individual of *Ostrea edulis* showing a clear breakage at the shell extremity (black arrow), resulting from an attempted predation by *Callinectes sapidus*.

**Figure 5 animals-15-03553-f005:**
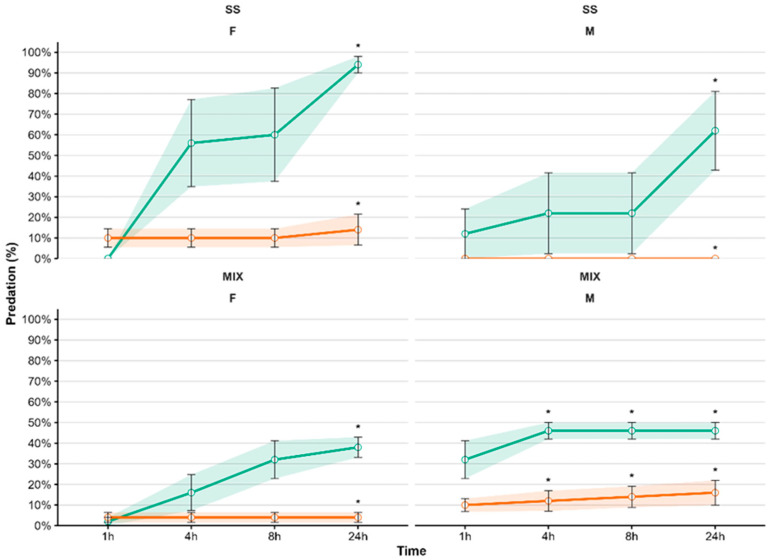
Predation by female (F) and male (M) *Callinectes sapidus* on small individuals of *Ostrea edulis* (green line) and *Paracentrotus lividus* (orange line), offered either as single prey species (SS) or a mix (MIX) of the two species. No predation was recorded on medium and large individuals of *O. edulis* and *P. lividus*; therefore. Asterisks indicate significant differences between species. Values are expressed as mean ± SE (*n* = 10).

**Figure 6 animals-15-03553-f006:**
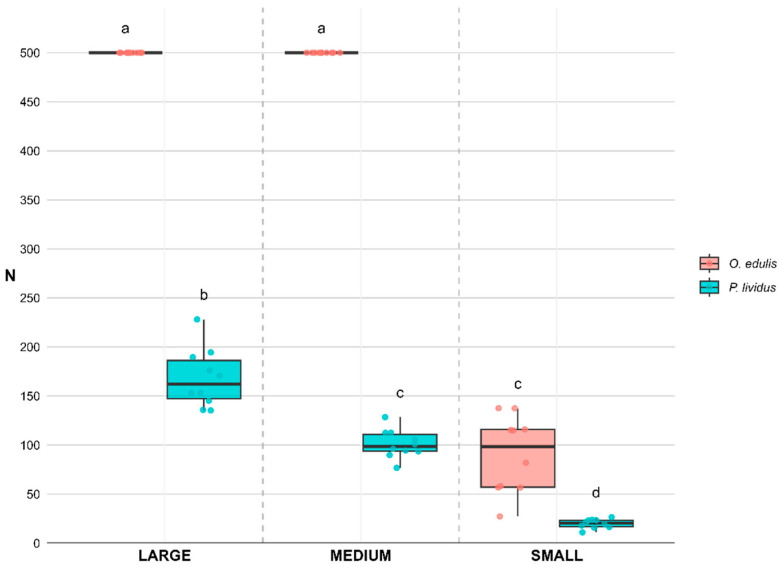
Compressive force (newtons, N) required to crush prey species (*Ostrea edulis* and *Paracentrotus lividus*) and size (Large, Medium, and Small). Medium and large individuals of *O. edulis* did not crush under the maximum force applicable with the mechanical press (500 N). Superscript letters indicate significant differences among treatments. Values are expressed as mean ± SE (*n* = 10).

**Figure 7 animals-15-03553-f007:**
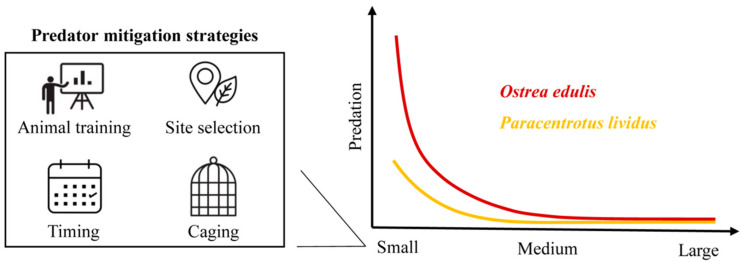
Conceptual representation of predator mitigation strategies and the relationship between prey size and predation by *Callinectes sapidus* on *Ostrea edulis* and *Paracentrotus lividus*. Management strategies such as animal training (pre-release conditioning), site selection, timing of release, and caging can be adopted to reduce predation pressure on small-sized individuals and improve the success of restoration programs.

**Table 1 animals-15-03553-t001:** Biometrics of predators (*Callinectes sapidus*, *n* = 5 per sex) and prey (*Ostrea edulis* and *Paracentrotus lividus*, *n* = 150 per species). Wet weight was measured to the nearest 0.1 g and size to the nearest mm. Values are expressed as mean ± SE.

**PREDATOR**	**Biomass (g)**	**Length (mm)**	**Width (mm)**	**Claw (mm)**
*C. sapidus* male	296.5 ± 20.5	75.4 ± 1.4	152 ± 2.8	95.7 ± 1.9
*C. sapidus* female	188.8 ± 15.7	70.4 ± 2.5	145 ± 7.1	66.8 ± 1.1
**PREY**	**Biomass (g)**	**Length (mm)**	**Diameter (mm)**	**Height (mm)**
*O. edulis*—large	123.4 ± 2.4	82.7 ± 0.7		33 ± 0.9
*O. edulis*—medium	31.1 ± 0.5	60.6 ± 0.4		15.2 ± 0.2
*O. edulis*—small	3.2 ± 0.1	32.9 ± 0.3		7.1 ± 0.2
*P. lividus*—large	62.2 ± 1.1		53.3 ± 0.4	33.4 ± 0.4
*P. lividus*—medium	14 ± 0.4		31.3 ± 0.3	19.1 ± 0.4
*P. lividus*—small	1.1 ± 0.1		13.5 ± 0.1	7.9 ± 0.1

**Table 2 animals-15-03553-t002:** Results of the *t*-test and Kruskal–Wallis test assessing differences between the sex of predator (Ma = male; Fe = female) and among size classes (S = small; M = medium; L = large) of *Ostrea edulis* and *Paracentrotus lividus*, respectively. Significant results (*p* < 0.05) are indicated in bold.

** *Callinectes sapidus* **	F	*p*	
Biomass	188.800	**0.0031**	Ma > Fe
Length	70.400	0.1155	Ma = Fe
Width	145.000	0.3871	Ma = Fe
Claw	67.800	**0.0001**	Ma > Fe
** *Ostrea edulis* **	H _2, 90_	*p*	
Biomass	79.12153	**0.0000**	L > M > S
Length	79.21347	**0.0000**	L > M > S
Height	79.39078	**0.0000**	L > M > S
** *Paracentrotus lividus* **	H _2, 270_	*p*	
Biomass	239.1183	**0.000**	L > M > S
Diameter	239.6507	**0.000**	L > M > S
Height	238.7411	**0.000**	L > M > S

**Table 3 animals-15-03553-t003:** Results of General Liner Model test for *Callinectes sapidus* predation at 1, 4, 8 and 24 h of exposure, in relation to the sex of predator (Sex; Ma = male, Fe = female), prey species (Species; Oe = *Ostrea edulis*, Pl = *Paracentrotus lividus*) and prey size (S = small; M = medium; L = large), when offered as single prey species. Significant results at *p* < 0.05 are indicated in bold.

		1 h			4 h			8 h			24 h		
Effect	df	MS	F	*p*	MS	F	*p*	MS	F	*p*	MS	F	*p*
Intercept	1	0.0657	4.5904	0.0373	0.5389	14.6628	0.0004	0.5591	14.6769	0.0004	1.5708	74.4895	0.0001
Sex	1	0.0032	0.2215	0.6401	0.1554	4.2290	**0.0452**	0.1664	4.3672	**0.0420**	0.1279	6.0639	**0.0174**
Species	1	0.0032	0.2215	0.6401	0.1777	4.8341	**0.0328**	0.1893	4.9704	**0.0305**	0.7973	37.8073	**0.0001**
Size	2	0.0657	4.5904	**0.0150**	0.5389	14.6628	**0.0001**	0.5591	14.6769	**0.0001**	1.5708	74.4895	**0.0001**
Sex-Species	1	0.0657	4.5904	**0.0373**	0.0067	0.1814	0.6721	0.0091	0.2383	0.6276	0.0001	0.0004	0.9846
Sex-Size	2	0.0032	0.2215	0.8022	0.1554	4.2290	**0.0203**	0.1664	4.3672	**0.0181**	0.1279	6.0639	**0.0045**
Species-Size	2	0.0032	0.2215	0.8022	0.1777	4.8341	**0.0122**	0.1893	4.9704	**0.0109**	0.7973	37.8073	**0.0001**
Sex-Species-Size	2	0.0657	4.5904	**0.0150**	0.0067	0.1814	0.8347	0.0091	0.2383	0.7889	0.0001	0.0004	0.9996
Error	48	0.0143			0.0368			0.0381			0.0211		
**Tukey’s HSD**		S > M = L			Fe > Ma			Fe > Ma			Fe > Ma		
					Oe > Pl			Oe > Pl			Oe > Pl		
					S > M = L			S > M = L			S > M = L		
					S(Fe) > S(Ma) = M = L			S(Fe) > S(Ma) = M = L			S(Fe) > S(Ma) > M = L		
					S(Oe) > S(Pl) = M = L			S(Oe) > S(Pl) = M = L			S(Oe) > S(Pl) = M = L		

**Table 4 animals-15-03553-t004:** Results of General Liner Model test for *Callinectes sapidus* predation at 1, 4, 8 and 24 h of exposure, in relation to the sex of predator (Sex; Ma = male, Fe = female), prey species (Species; Oe = *Ostrea edulis*, Pl = *Paracentrotus lividus*) and prey size (S = small; M = medium; L = large), when offered as a mix of the two prey species. Significant results at *p* < 0.05 are indicated in bold.

		1 h			4 h			8 h			24 h		
Effect	df	MS	F	*p*	MS	F	*p*	MS	F	*p*	MS	F	*p*
Intercept	1	0.7855	28.9739	0.0001	1.7092	75.3869	0.0001	2.0878	74.4937	0.0001	2.4438	131.4860	0.0001
Sex	1	0.2914	10.7505	**0.0019**	0.2226	9.8189	**0.0029**	0.1460	5.2097	**0.0269**	0.0903	4.8573	**0.0324**
Species	1	0.0212	0.7821	0.3809	0.2809	12.3905	**0.0009**	0.3841	13.7042	**0.0005**	0.4919	26.4686	**0.0001**
Size	2	0.7855	28.9739	**0.0001**	1.7092	75.3869	**0.0001**	2.0878	74.4937	**0.0001**	2.4438	131.4860	**0.0001**
Sex-Species	1	0.0682	2.5144	0.1194	0.0244	1.0783	0.3043	0.0003	0.0126	0.9110	0.0099	0.5331	0.4689
Sex-Size	2	0.2914	10.7505	**0.0001**	0.2226	9.8189	**0.0003**	0.1460	5.2097	**0.0090**	0.0903	4.8573	**0.0120**
Species-Size	2	0.0212	0.7821	0.4632	0.2809	12.3905	**0.0001**	0.3841	13.7042	**0.0001**	0.4919	26.4686	**0.0001**
Sex-Species-Size	2	0.0682	2.5144	0.0915	0.0244	1.0783	0.3483	0.0003	0.0126	0.9874	0.0099	0.5331	0.5902
Error	48	0.0271			0.0227			0.0280			0.0186		
**Tukey’s HSD**		Ma > Fe			Ma > Fe			Ma > Fe			Ma > Fe		
		S > M = L			Oe > Pl			Oe > Pl			Oe > Pl		
		S(Ma) > S(Fe) = M = L			S > M = L			S > M = L			S > M = L		
					S(Ma) > S(Fe) > M = L			S(Ma) > S(Fe) > M = L			S(Ma) > S(Fe) > M = L		
					S(Oe) > S(Pl) > M = L			S(Oe) > S(Pl) > M = L			S(Oe) > S(Pl) > M = L		

## Data Availability

The original contributions presented in this study are included in the article. Further inquiries can be directed to the corresponding author.

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
