# Peer review of "The Invasive Alien Species Callinectes sapidus Threatens the Restoration of Ostrea edulis and Paracentrotus lividus in the Mediterranean Sea"

_animals, 2025, doi:10.3390/ani15243553_

Round 1

Reviewer 1 Report

Comments and Suggestions for Authors

The invasive alien species Callinectes sapidus threatens the restoration of Ostrea edulis and Paracentrotus lividus in the Mediterranean Sea Brundu et al 

The research is based on laboratory work to understand the predation of C. sapidus to three different sizes of Ostrea edulis and Paracentrotus lividus in the lab.

The authors explained the history of Callinectes sapidus and their interaction with other animals in the Mediterranean after their invasion.

They tested both male and female C. sapidus on different size classes of O. edulis and P. lividus under controlled laboratory conditions.

Crabs were offered single or mixed prey species, and consumption was monitored over 24 h. Small oysters were heavily preyed upon (>62%), whereas medium (~60 mm) and large (~82 mm) individuals were not consumed.  In contrast, P. lividus was only occasionally consumed (<32%) at the smallest size (~13.5 mm diameter).

Below are my comments.

The research design presented well in Figure 1 and 2 and relevant tables but statistical analyses of Mann whitney U test and Kruskall wallis tests may not be enough to show the different factors, gender of crab, prey species, prey size etc. I think General lineer model would be ideal to use to show such multiple factors and their differences among themselves via Tukey (as this was given in figure 6). This data can only be obtained via such comparison. I therefore suggest authors to chaeck the statistical results and degrees of freedom (df) which commonly used. The significance levels is very low in tables but usually given as P<0.05 only.

Line 356, The reference (43) and Preston et al 2020 used twice in this sentence. Line 55-56, country names and the relevant study has to be checked and country names and their study sites could be useful to present here. I therefore the authors should check all the references cited correctly.

Author Response

The research design presented well in Figure 1 and 2 and relevant tables but statistical analyses of Mann whitney U test and Kruskall wallis tests may not be enough to show the different factors, gender of crab, prey species, prey size etc. I think General lineer model would be ideal to use to show such multiple factors and their differences among themselves via Tukey (as this was given in figure 6). This data can only be obtained via such comparison. I therefore suggest authors to chaeck the statistical results and degrees of freedom (df) which commonly used. The significance levels is very low in tables but usually given as P<0.05 only.

Done. We changed the statistical analysis by adopting a General Linear Model test.

Line 356, The reference (43) and Preston et al 2020 used twice in this sentence. Line 55-56, country names and the relevant study has to be checked and country names and their study sites could be useful to present here. I therefore the authors should check all the references cited correctly.

Done. We deleted Preston et al. 2020 as it was used twice in the sentence and we checked the relevant studies and references cited in lines55-56.

Reviewer 2 Report

Comments and Suggestions for Authors

Review of the manuscript 'The invasive alien species Callinectes sapidus threatens the restoration of Ostrea edulis and Paracentrotus lividus in the Mediterranean Sea' by Gianni Brundu, Philip Graham, Mattia Corrias, Cheoma Frongia, and Stefano Carboni.

This study investigates the threat posed by the invasive blue crab, Callinectes sapidus, to restoration efforts for the native European flat oyster (Ostrea edulis) and purple sea urchin (Paracentrotus lividus) in the Mediterranean Sea. Through laboratory experiments, it was found that blue crabs pose a major risk to small, juvenile oysters, with predation rates exceeding 60-90%, while largely ignoring larger oysters and sea urchins. The crabs showed a strong preference for oysters over sea urchins. The study concludes that the release of juvenile oysters smaller than 60 mm is highly vulnerable to this new predator. To ensure restoration success, the authors recommend strategic management, such as releasing larger oysters, timing deployments during the crab's winter dormancy, and using protective caging to mitigate predation pressure.

Introduction

Pg 2 Ln 47-50: The logical link between 'trophic cascading effects' and the subsequent 'importance to manage' is unclear. The authors should more explicitly state why these cascading effects make management imperative.

Pg 2 Ln 68-69: To strengthen the argument for alternative strategies like restocking, the authors should clarify the most important reasons why these measures are 'often ineffective'.

Pg 2 Ln 85-86: The authors should provide some data on the abundance dynamics of Callinectes sapidus to illustrate the mentioned pattern more rigorously.

Material and methods

Pg 4 Ln 131-132: The authors should report the depth range there the crabs were collected. They should report some data on environmental conditions of the study area and, if available, data on the abundance of Callinectes sapidus at this site. Did the authors determine the shell condition of experimental crabs? Was the shell condition same in these specimens?

Pg 5 Ln 151-153: The authors should mention the cultivation conditions (food, temperature) for the oysters during the farming period in the lagoon, as this could influence shell strength.

Pg 5 Ln 175-177:  To allow for verification of the force calculation, the authors should report the technical specifications of the custom-built press, such as the piston area.

Did the authors monitor the health status and mortality of the experimental crabs during the acclimation and experimental period?

Results

Pg 6 Ln 194-201: This text and table 2 should be updated with comparisons between male and female crabs.

Pg 6 Ln 205-206: The authors should mention whether these 'predation attempts' were quantified in any way as it can provide valuable context on crab behavior.

Discussion

Pg 10 Ln 272-273: 'Crabs appeared disinterested' is not a good explanation for the supposed poor palatability. The authors should explain the basis for this hypothesis based on a more rigorous reason such as chemical or morphological parameters of P. lividus.

Pg 11 Ln 290-293: The authors should explain the potential reasons for this contrast more thoroughly. Could crab size or methodology also play a role?

Pg 11 Ln 311-312: The authors should explain these contrasting patters. What ecological or morphological factors might cause a predator to prefer larger prey in one species and smaller in another?

Pg 12 Ln 363-365: The authors should justify this recommendation more strongly. They should state if their experiments were conducted at temperatures simulating November conditions, and if the growth conditions for the species are still adequate in late autumn.

Author Response

Introduction

Pg 2 Ln 47-50: The logical link between 'trophic cascading effects' and the subsequent 'importance to manage' is unclear. The authors should more explicitly state why these cascading effects make management imperative.

We agree and thank the reviewer for the suggestion. We rephrased more explicitly state why these cascading effects make management imperative.

 Pg 2 Ln 68-69: To strengthen the argument for alternative strategies like restocking, the authors should clarify the most important reasons why these measures are 'often ineffective'.

We agree with the reviewer and we clarified the most important reasons why these measures are 'often ineffective'.

 Pg 2 Ln 85-86: The authors should provide some data on the abundance dynamics of Callinectes sapidus to illustrate the mentioned pattern more rigorously.

Done. We added some data on the abundance dynamics of C. sapidus in the Mediterranean Sea (lines 95-100).

 Materials and methods

Pg 4 Ln 131-132: The authors should report the depth range there the crabs were collected. They should report some data on environmental conditions of the study area and, if available, data on the abundance of Callinectes sapidus at this site. Did the authors determine the shell condition of experimental crabs? Was the shell condition same in these specimens?

Done. As suggested by the reviewer we added the depth the crabs were collected, some environmental conditions of the lagoon, and we explicit the shell condition of experimental crabs. Unfortunately, data on the abundance of C. sapidus at this site are not available.

 Pg 5 Ln 151-153: The authors should mention the cultivation conditions (food, temperature) for the oysters during the farming period in the lagoon, as this could influence shell strength.

Done. As suggested by the reviewer we mentioned the farming methods and conditions (food, temperature).

 Pg 5 Ln 175-177:  To allow for verification of the force calculation, the authors should report the technical specifications of the custom-built press, such as the piston area.

Done, we added the piston area of the custom-built press.

 Did the authors monitor the health status and mortality of the experimental crabs during the acclimation and experimental period?

Done. As described in the previous comment, we added the  monitoring of shell condition of experimental crabs and their health status (lines 146-150).

 Results

Pg 6 Ln 194-201: This text and table 2 should be updated with comparisons between male and female crabs.

Done. We added text and table 2 with comparison between male and female crabs (lines 218-222).

 Pg 6 Ln 205-206: The authors should mention whether these 'predation attempts' were quantified in any way as it can provide valuable context on crab behavior.

Done. As suggested by the reviewer we mention that the “predation attempts” were not quantified during the experiment.

 Discussion

Pg 10 Ln 272-273: 'Crabs appeared disinterested' is not a good explanation for the supposed poor palatability. The authors should explain the basis for this hypothesis based on a more rigorous reason such as chemical or morphological parameters of P. lividus.

Done. As suggested by the reviewer we rephrased explaining our hypothesis on a low predation (lines 297-299).

 Pg 11 Ln 290-293: The authors should explain the potential reasons for this contrast more thoroughly. Could crab size or methodology also play a role?

We agree with the reviewer. We modified the text clarifying no play role of crab size and methodology (lines 326-328).

 Pg 11 Ln 311-312: The authors should explain these contrasting patters. What ecological or morphological factors might cause a predator to prefer larger prey in one species and smaller in another?

We agree with the reviewer. We modified the text explaining what What ecological or morphological factors might cause a predator to prefer larger prey (lines 347-349).

 Pg 12 Ln 363-365: The authors should justify this recommendation more strongly. They should state if their experiments were conducted at temperatures simulating November conditions, and if the growth conditions for the species are still adequate in late autumn.

We agree with the reviewer. We modified the text explicating the environmental conditions in November, as well as the potential needing to acclimate the animals prior to their release at different climatic conditions (lines 402-407).